# The Influence of Shallow Peatland Water Quality on Characteristics of the Occurrence of Selected Herb Species in the Peatlands of Eastern Poland

**DOI:** 10.3390/ijerph20042788

**Published:** 2023-02-04

**Authors:** Artur Serafin, Magdalena Pogorzelec, Urszula Bronowicka-Mielniczuk

**Affiliations:** 1Department of Environmental Engineering and Geodesy, University of Life Sciences in Lublin, Leszczyńskiego 7, 20-069 Lublin, Poland; 2Department of Hydrobiology and Protection of Ecosystems, University of Life Sciences in Lublin, B. Dobrzańskiego 37, 20-262 Lublin, Poland; 3Department of Applied Mathematics and Computer Science, University of Life Sciences in Lublin, Głęboka 28, 20-612 Lublin, Poland

**Keywords:** herbs, water hydrochemistry, physicochemical factors, habitat

## Abstract

The aim of the analysis was to compare the physicochemical variables of the quality of shallow groundwater in the peatlands of Eastern Poland in the context of the occurrence of selected herb species with similar habitat requirements: bogbean (*Menyanthes trifoliata*), small cranberry (*Oxycoccus palustris*), and purple marshlocks (*Comarum palustre*). The analysis of the quality variables of the shallow groundwater included the following physicochemical variables: reaction (pH), electrolytic conductivity (EC), dissolved organic carbon (DOC), total nitrogen (N_tot._), ammonium nitrogen (N-NH_4_), nitrite nitrogen (N-NO_2_), nitrate nitrogen (N-NO_3_), total phosphorus (P_tot._), phosphates (P-PO_4_), sulphates (SO_2_), sodium (Na), potassium (K), calcium (Ca), and magnesium (Mg). Internal metabolism was shown to influence the hydro-chemical status of peatland water, free of substantial human impact. The variables tested were within the range of the habitat preferences of the herb species and indicated that they have a wide ecological tolerance. However, their identical habitat preferences were not reflected in identical values for the physicochemical variables of the water essential for building populations of these species. The occurrence of these plant species was also shown to be determined by the hydro-chemical characteristics of the habitat, but the characteristics of their occurrence did not indicate the hydro-chemical aspect of the habitat.

## 1. Introduction

The hydro-chemical characteristics of surface water and groundwater are determined by physic-geographical and climatic factors, as well as by anthropogenic determinants associated with the type of land management [1,2,3]. Geological and topographic features determine surface runoff and landscape retention, while climate characteristics affect temperature, humidity, atmospheric pressure, and the chemical composition and distribution of precipitation. The means and intensity of human management, and use of a given area, influence the local and regional water balance [4,5,6,7]. All of these elements influence the chemical composition of groundwater; they are reflected in the species composition and life forms of the flora and in the diversity of plant communities, and they determine the biocoenotic stability of a given community [3,8,9].

However, plant species face an adaptation problem due to climate disturbances [10,11,12] and increased human impact on the environment [3,13]. This leads to a number of habitat changes, leading to contraction of the natural sites of the plants.

In Poland, progressive changes in atmospheric circulation and solar radiation have been observed since the mid-twentieth century, resulting in temperature fluctuations at various times of the year and variable distribution and intensity of precipitation. The eastern part of the country is also affected by these changes [14,15].

Direct and indirect anthropogenic transformations associated with industry, agriculture, municipal engineering, transport, tourism, and recreation affect all aspects of the environment [16]. For hydrological status, apart from the influx of pollutants, the most important factors are disturbances of the water regime, changes in the rate of nutrient accumulation, and fluctuations in the groundwater table [3].

Natural and anthropogenic changes in the environment result in the degradation and fragmentation of habitats for many rare plant species [6,7,17,18] and potentially valuable and useful herb species.

The aim of the present analysis was to compare the physicochemical variables of the quality of shallow groundwater in the peatlands of Eastern Poland in the context of the occurrence of selected herb species with similar habitat requirements, i.e., bogbean (*Menyanthes trifoliata*), small cranberry (*Oxycoccus palustris*), and purple marshlocks (*Comarum palustre*). The analysis was based on the results of previous research on the hydro-chemical characteristics favoring the occurrence of a given herb species [3,18,19] using a methodology developed for ecological analysis of peatland ecosystems [9].

The overriding goal of the habitat analyses was to establish a range of values for each physicochemical variable of the groundwater within which a given species can carry out its life strategy, i.e., its ecological tolerance (hydro-chemical characteristics of habitats in the Łęczyńsko-Włodawskie Lake District).

The following research hypothesis was adopted for the study:Peatland groundwater quality determines the occurrence of peatland species of herbs in different ways.

Based on this hypothesis, specific research problems were formulated as questions:
In conditions assumed to be natural (in protected natural areas), can the effect of human impact be observed in the hydro-chemical characteristics of habitats?Are the hydro-chemical characteristics of the sites of plant species occurrence a reflection of their habitat preferences?If the habitat characteristics associated with the preferences of a given species are maintained, are there hydro-chemical factors differentiating their occurrence?Can the characteristics of the occurrence of a given plant species be used as an indicator of progressive changes in habitat conditions?

The practical aim of the study was to establish ranges of values of physicochemical variables of groundwater for experimental organic cultivation of selected herb species in optimal hydro-chemical conditions. The long-term goal is to create an herbal medicine with a natural composition of medicinal ingredients in the conditions of an organic agroecosystem.

## 2. Material and Methods

Purple marshlocks (*Comarum palustre* L.) is a perennial plant of the family Rosaceae. Its medicinal properties are valued mainly in Eastern Europe [7,20,21]. It is present in North America, Central, Northern Europe, Siberia, Greenland, and Iceland and is associated with shallow marshes, peatlands, and acidic wetlands [7,22,23]. In Poland, it grows at many sites and is acquired from natural habitats. In recent decades, however, due to hydrological changes, there has been a decline in the number of sites and a marked decline in the number of individuals at these sites [7,24,25]. Rhizomes and aerial shoots obtained from spring to autumn are used for pharmaceutical purposes [7,21,26,27,28,29].

Bogbean (*Menyanthes trifoliata* L.) is an herbal plant species acquired in Poland only from natural habitats. It is a perennial plant of the family Menyanthaceae, with a thick cylindrical rhizome [19,30,31]. It is native to the Northern Hemisphere and temperate circumpolar zone of Europe, Asia, and North America. It grows on the edges of marshes and transitional and raised bogs, as well as in acidic wet sedge meadows [19,31]. The herbal material is the leaves, which are obtained during flowering [19,32,33,34,35]. Changes in the water balance in many parts of Poland are leading to a gradual decline in both the number of populations and the number of individuals in them [19,24,31]. For this reason, the species is under partial protection in Poland (Regulation of the Minister of Environment, 20 January 2012).

Small cranberry (*Oxycoccus palustris* Pers.), one of the most valuable herbal plants, is also usually obtained from nature. It is a shrub in the family Ericaceae, with delicate, creeping shoots up to 100 cm high. It is found in Northern and Central Europe, North Asia, and North America; in Poland it grows in wetlands associated with ombrogenous peatlands [18,35,36,37]. The herbal material is the fruit—a spherical, multi-seeded, juicy, red berry obtained in autumn [18,38,39,40]. 

In terms of life strategy, in the peatlands of Eastern Poland, these herb species prefer the same habitat conditions. As chamaephytes and hydrophytes, they are found in cold habitats with organogenic humus soils characteristic of raised or transitional oligotrophic or mesotrophic bogs, but rich in organic matter, wet, acidic or moderately acidic, with moderate light, which describes the climate of the Circumboreal and Holarctic regions [3,18,19,24,35].

### Study Area and Methods

The locations selected for the ecological analysis in the peatlands of Eastern Poland (Łęczyńsko-Włodawskie Lake District) were representative natural and semi-natural sites with potentially minimal human impact and regulated hydrological conditions (Poleski National Park and its buffer zone). They were lake-peat bog complexes, i.e., Lake Bikcze (B), Lake Karaśne (K), Lake Długie (D), and Lake Moszne (M); the mid-forest bog Blizionki (BZ); and the Dekowina wilderness area (DK) (Figure 1).

The criterion for selection of specific sampling sites at the locations was the abundance of the plant species chosen for analysis. The threshold of abundance was 10% area cover by a given herb species.

A representative 100 m^2^ (10 m × 10 m) sampling plot was arbitrarily designated at each location (a patch of vegetation adequate to the location, determined on the basis of species composition and the percentage share of individual plant species in the phytocenosis). In the center of each plot a piezometer was installed for sampling of shallow peat bog water (a perforated PVC pipe 1 m in length and 10 cm in diameter, closed with a plug for a total of six piezometers). The water samples for laboratory analysis were collected seven times a year during the growing season in the years 2011–2013 according to the method adopted for habitat analysis [9]. To reflect the range of natural conditions, samples were taken in a variety of weather conditions.

Qualitative analysis of the water samples was performed at the Central Agroecological Laboratory (CLA) of the University of Life Sciences in Lublin using certified methods in accordance with standard procedures. The samples were analyzed for 14 physicochemical variables of importance for the functioning of any phytocoenosis: pH by the potentiometric method (PN-EN ISO 10523); electrolytic conductivity (EC) by conductometry (PN-EN ISO 27888); amount of dissolved organic carbon (DOC) by spectrometry (PN-EN 1484); content of nitrogen fractions, i.e., total nitrogen (N_tot_) by flow analysis (ISO 29441), ammonium nitrogen (NH_4_) by flow analysis (PN-EN ISO 11732), nitrates (NO_3_), and nitrites (NO_2_) by flow analysis (PN-EN ISO 13395); content of phosphorus fractions, i.e., total phosphorus (P_tot_) and phosphate phosphorus (PO_4_) by spectrophotometry (PN-EN ISO 6878); concentration of sulphates (SO_4_) by ion-exchange chromatography (PN-EN ISO 10304-1); and base cations by atomic absorption spectrometry (PN-EN ISO 9964-2 for potassium (K) and sodium (Na), PN-EN ISO 7980 for calcium (Ca) and magnesium (Mg)). The factors selected for the qualitative assessment of the habitat are consistent with the direction of possible regional environmental influences (agriculture, unorganized tourism) and the character of the habitat preference of herbs.

Due to the lack of normal distribution and the heterogeneity of variance in the data sets, the non-parametric Mann–Whitney U test was used to compare the distributions of values of properties in individual habitats. The analysis was performed for a significance level of 0.05 and for two categories of study sites: abundant and not abundant in individuals of a given species. Ranges of values for the environmental factors together with the statistical analyses were presented in the form of graphs showing the properties of the empirical distributions for all species combined.

For a comparative analysis, graphs were also used to present the range of values of physicochemical factors collectively for the sites with the highest and lowest abundance of a given species. The Pearson correlation between physicochemical variables at the study sites were determined as well, and the results were presented in correlograms. Next, principal component analysis was performed for P_tot_, N_tot_, DOC, EC, and pH (the most important variables for evaluation of habitat status), and the results were presented on an ordination biplot.

All statistical analyses were performed in the open-source software R, version 4.1.0 [41] using the following packages: corrplot (ver. 0.9), ggplot2 (ver. 3.3.5), factoextra (ver. 1.0.7), and stats (ver. 4.1.0).

## 3. Results

In the initial stages of the research, the abundance of individuals of the studied plant species in selected locations was determined based on their percentage share in the phytocenosis in accordance with the adopted principle: abundant > 10% ≥ not abundant. Abundance of each species at the various locations was ranked as follows: for bogbean B = D = DK (most abundant) > M > K = BZ (least abundant); for small cranberry M = K = D = DK (most abundant) > B = BZ (least abundant); and for purple marshlocks M = B = D (most abundant) > K = BZ = DK (least abundant). For *Menyanthes trifoliata*, sites abundant in individuals were characterized by a 20% share, not abundant sites with a 5–10% share for *Oxycoccus palustris*, respectively, 30% for abundant sites and 5% for less abundant sites, and for *Comarum palustre*, 20% share in abundant sites and 10% share in phytocoenoses of less abundant sites.

In the first stage of the research, the empirical distribution of the variables was analyzed for pooled data. Variation in the values of these factors determines the range of ecological tolerance of individuals of the herb species in relation to the variables themselves and to the sites of occurrence of the species (Figure 2).

For some of the physicochemical variables (K, Ca, nitrogen fractions, and P-PO_4_), single substantial deviations were noted, as evidenced by greater divergence of the means relative to the median. The median values for all variables were below the means (Figure 2). Factors showing relatively minor variation for all locations of the herb species were Mg, DOC, pH, Na, P_tot_, and SO_2_ (Figure 2).

More detailed information is provided by the comparative analysis of the values of the physicochemical variables of the peatland water for the sites with higher and lower abundance of a given species (Figure 3, Figure 4 and Figure 5). For these species, while their habitat preferences were the same, substantial differences were sometimes noted in this analysis.

For *Comarum palustre*, higher means were noted for all nitrogen fractions for the sites with lower abundance of the species (Figure 3). A similar trend was noted for Ca and Mg ions, pH, EC, and DOC, while for phosphorus fractions, Na and K ions, and SO_2_ there were no substantial differences between means for the two types of habitats (Figure 4 and Figure 5).

In the case of *Menyanthes trifoliata*, the distribution of means for the variables in relation to the abundance criterion was somewhat different. Higher means for the variables at the sites with higher abundance of the species were noted for K, DOC, N_tot_, N-NO_2_, N-NH_4_, and P_tot_. Higher means for Ca, Mg, EC, and pH were recorded for the sites with lower abundance in the species, while the values for Na, N-NO_3_, P-PO_4_, and SO_2_ were similar for both habitat types (Figure 3, Figure 4 and Figure 5).

In the analysis of the occurrence of *Oxycoccus palustris*, higher mean values for the sites with higher abundance were recorded for all nitrogen fractions, Ca, and pH; similar values for all sites for SO_2_ and DOC; and higher means for Na, K, Mg, and P fractions for the sites with low abundance (Figure 3, Figure 4 and Figure 5).

The correlation matrices for the physicochemical variables determined for the study sites using the abundance criterion showed strong or moderate positive and negative correlations depending on the species (Figure 6, Figure 7 and Figure 8).

In the case of *Comarum palustre*, strong positive correlations were found at the abundant sites for N-NH_4_ with P_tot_, P-PO_4_, and K; P_tot_ with P-PO_4_ and K; and the pairs P-PO_4_–K, Mg–Na, Mg–K, and EC–Ca, while strong negative correlations were obtained for pH with N-NH_4_, P_tot_, and P-PO_4_. At sites with low abundance, strong positive correlations were shown for P_tot_–P-PO_4_, P_tot_–Na, N-NO_2_–SO_2_, N-NO_2_–Mg, N-NH_4_–N_tot_., N-NO_3_–Mg, SO_2_–Mg, Ca–Mg, EC–Ca, EC–pH, P-PO_4_–Na, and pH–Ca. The strong negative correlations were shown between N_tot_.–EC, N_tot_.–pH, N-NO_3_–DOC, N-NH_4_–EC, N-NH_4_–pH, Ca–DOC, DOC–EC, and DOC–pH (Figure 6).

In the case of *Menyanthes trifoliata*, strong positive correlations for the abundant sites were found for N-NO_2_ with N-NO_3_, SO_2_, and Ca; N-NO_3_ with SO_2_, Ca and Mg; SO_2_ with Ca and Mg; and also for P_tot_–P-PO_4_, N-NH_4_–DOC, Na–K, K–DOC, and Ca–Mg. Strong negative correlations at these sites were noted for pH with P fractions, K, and DOC. For sites with low abundance, only strong positive correlations were shown. These were for P_tot_ with P-PO_4_ and Na; Ca with Mg, EC, and pH; and the pairs pH–EC and N-NO_2_–SO_2_ (Figure 7). For *Oxycoccus palustris* at abundant sites, strong positive correlations were noted for N_tot_ with N-NH_4_, P_tot_, K, and DOC; for Ca with Mg, EC, and pH; and for the pairs N-NO_2_–SO_2_; P_tot_–K; Na–K, and EC–pH. For the same sites, negative correlations were found for P_tot_ and DOC with pH and EC. At sites with low abundance, strong positive correlations were shown for N-NO_2_ with Na and Ca; N-NO_3_ with N-NH_4_ and P_tot_; N-NH_4_ with P-PO_4_ and K; P-PO_4_ with P_tot_ and Mg; Ca with Mg, Na, and pH; and EC with pH. Strong negative correlations at these sites were obtained for pH with K and N-NH_4_ (Figure 8).

The Mann–Whitney U test at a significance level of 0.05 revealed significant differences for some of the physicochemical variables in relation to the study sites. In the case of *Comarum palustre*, such differences depending on the plant’s abundance at the site were noted for N-NH_4_, DOC, Ca, and EC (Table 1).

For *Menyanthes trifoliata*, significant differences were obtained for N_tot_, Ca, pH, and EC, and in the case of *Oxycoccus palustris*, for P_tot_ and P-PO_4_ (Table 1). The distribution of the remaining physicochemical factors did not show statistically significant differences between the sites with greater or lower abundance of a given species.

The first two principal component (PC) axes accounted for 91.1% of the explained variation (PC1 = 61.1%; PC2 = 24.5%). The first principal component (PC1) had strong positive associations with N_tot_ and negative associations with pH (Figure 9).

The second component (PC2) had strong positive associations with EC, DOC, and P_tot_. Furthermore, two pairs of strongly positively correlated variables can be distinguished: P_tot_–DOC and EC–pH, and they have almost no correlation with each other. Regarding the study sites, the values of the physical-chemical variables at locations B and DK were similar (N_tot_ values were higher than average, while pH values were lower than average). At sites D and M, the N_tot_ and pH levels were similar (below average), while at sites K and BZ the values of pH and EC were similar (Figure 9).

## 4. Discussion

Wetland ecosystems, including peatlands, are systematically declining in area in most European countries due to hydrotechnical management [42,43]. The reduction in peatland area limits flood water retention, groundwater recharge, organic carbon accumulation, and biological nutrient retention, and most importantly, leads to the destruction of habitats of valuable plant and animal species [44]. Disturbances of the hydrological conditions of peatlands leading to dehydration of peat deposits also affects the hydro-chemical status of the habitat, causing numerous biocoenotic changes including a loss of biodiversity [3,6,18,19,45].

An additional consequence of human impact is pollution of peatland water, to which industry, agriculture, municipal engineering, transport, tourism, and recreation are major contributors [16,46].

In peatland ecosystems the proximity of agricultural land is the main source of biogenic substances and post-production pollutants, accelerating eutrophication of their habitats [9,47,48]. Intensive, industrialized agriculture also alters the structure of the landscape itself (fragmentation of natural habitats) and, for this reason, is ranked first in global assessments of the causes of impoverishment of nature [9,49]. Groundwater quality is also strongly associated with rural settlement (point sources of pollution), and is responsible for the highest concentrations of organic pollutants, ammonium nitrogen (N-NH_4_), total phosphorus (P_tot_), phosphates (P-PO_4_), and chlorides (Cl). In addition, rural areas are considered to have the greatest impact on the quality of surface water [5,7,50].

The peatland ecosystems of Eastern Poland, despite their exceptionally valuable natural and landscape features, are also subject to the impact of human activity. This includes hydrotechnical reconstruction as well as the direct effects of agriculture, the mining industry, tourism, and recreation.

In the peatland ecosystems of Eastern Poland, drainage projects have shaped hydrological conditions and the nutrient balance of the entire Łęczyńsko-Włodawskie Lake District since the mid-nineteenth century [51]. However, they were only of local importance, in contrast to the construction (1954–1961) and operation of the Wieprz–Krzna drainage canal. The canal, together with several lakes converted to retention reservoirs, caused a reduction in the level of the groundwater table in the region, on average by 50–80 cm. In addition, water that is hydro-chemically foreign to the region flows through the canal, altering the habitat conditions of the peatlands in this area. Another anthropogenic factor causing water drainage is the operation of the Lublin Coal Basin, which further lowers the groundwater level throughout the region [52,53,54].

Taking into account possible anthropogenic impacts based on field reconnaissance studies supported by laboratory analyzes of shallow groundwater in peat bogs, representative locations were selected for various types of regional ecological studies in the Łęczyńsko-Włodawskie Lake District.

These were places with habitat naturalness confirmed by botanical research (species lists, preliminary phytosociological analysis and species similarity, values of factors within the anthropogenic changes in flora) with a potentially limited impact of anthropopressure and regulated hydrological conditions. These sites were used for a series of biocenotic analyses, including those concerning selected herb species. Comparing the research results to the repeatedly confirmed typological status of the habitat made it possible to determine the potential impact of anthropopressure on the locations selected for the study.

Hydro-chemical analysis of the habitats of selected herb species in the peatlands of Eastern Poland must, therefore, begin with a preliminary analysis of the hydrological conditions of the locations selected for the study. In this case, as it was not technically possible to perform reliable measurements of the changes in the position of the groundwater table, a subjective assessment was used: the availability of water in piezometers was checked when the samples were taken. On this basis, along with regional scientific information (Poleski National Park), it was established that the groundwater level in the peatlands was stable and similar to the peat level, showing only minor seasonal fluctuations with greater hydration following spring thaws and lower hydration in the summer (and thus this was not a limiting factor). All study sites were, therefore, characterized by significant hydration, whose degree was associated with the nature of the location itself (its connections with the close presence of a lake), and the time of year (thaws or changes in precipitation). Therefore, it was concluded that these sites had a high moisture level and regulated hydrological conditions which did not indicate disturbances associated with human impact.

The absence of dehydration of the peat deposits was confirmed by the analysis of the physicochemical variables of the shallow peatland water, as dehydration of biogenic sediment results in a high rate of mineralization of the organic matter contained in the surface layers of the soil. This process leads to the release of considerable amounts of biogenic substances, mainly nitrogen and sulfur, which entails acidification of hydrogenic soil through oxygenation of their compounds, as well as the loss of base cations due to biological retention by flora and leaching from the soil profile [55,56,57]. The reduction in pH initiates the gradual release of phosphorus by increasing the solubility of its compounds, e.g., apatite, strengite, and variscite, leading to activation of aluminum, which is toxic for plants [6,7,45,58]. An increased amount of chemically-reactive phosphorus in peatland soils entails the risk of its dispersion in the water resources, causing changes in the hydro-chemical characteristics and trophic status of the ecosystem [6,7,59].

These processes were not observed at the sites of the herb species selected for the study. The groundwater pH ranged from acidic to neutral, and the average ranges were slightly acidic (pH = 5.61–5.67). These are appropriate values for habitats of the transitional bogs and fens in the Łęczyńsko-Włodawskie Lake District [60] and are not indicative of significant human impact. It was established that biogenic substances in the peatland groundwater determined the abundance of the species in different ways despite their identical preferences, and indicated habitat eutrophication processes in different ways depending on the study site but without exceeding average mesotrophic values (according to [43]). The average N_tot_ concentration at the study sites varied widely from 16.92 to 45.31 mg·dm^−3^. Similarly, wide ranges were recorded for N-NH_4_ (0.55–0.76 mg·dm^−3^) and N-NO_2_ (0.06–4.33 mg·dm^−3^). N-NO_3_ did not exceed 0.2 mg·dm^−3^, and the average P _tot_ concentration ranged from 0.22 to 0.42 mg·dm^−3^. The possibility of secondary eutrophication of peatland ecosystems may also be determined by the content of sulfur compounds in the soil solution [57], as well as the concentrations of sodium, potassium, calcium, and magnesium—the main alkali elements determining the pH of soil solutions [61]. Their quantity is regulated by sorption or release from peatland sorption complexes [62]. The ranking of concentrations of alkali metals for all study sites and plant species was Ca > Na > K > Mg and was not indicative of any major disproportions in the rate and nature of release of elements from the soil sorption complexes, although it did demonstrate greater presence of calcium hydroxides in the soil solutions (a regional determinant). This is evidence of natural metabolic processes in the peatlands, as agricultural use of the catchment is conducive to an increase in concentrations of magnesium, calcium, and, to a small degree, sodium, and to a decrease in potassium content in its water bodies [63]. The high Ca:Mg ratio also confirms the lack of intensification of human impact as low values for such cation ratios indicate the effect of municipal pollution from developed areas [64]. The average concentrations of SO_4_ irons (2.51–4.94 mg·dm^−3^) and average EC values (99.28–184.8 µS·cm^−1^), which are an indirect measure of water mineralization and pollution, also did not suggest a significant degree of human impact at the study sites. Similarly, the average values for dissolved organic matter (DOC), which determines the accessibility of easily available forms of nitrogen and phosphorus associated with humic substances [65], did not indicate the possibility of secondary activation of biogenic compounds for plants.

During the laboratory analysis, however, occasional fluctuations in the physicochemical variables were noted, which can be linked to the dynamics of the internal metabolism of peatland ecosystems shaped by the simultaneous processes of organic matter mineralization and biological retention of nutrients by plant communities. These distinct processes are characterized by a labile balance dependent on varying metabolic activity of microorganisms, the primary production rate, and the complex of abiotic habitat factors varying in time and space [6,7,45]. Habitats of this hydro-chemical description are common in various types of peatlands in the Łęczyńsko-Włodawskie Lake District [60]; therefore, the locations selected for the study reflect the hydro-chemical composition of peatlands in the entire region [3,6,18,19]. No evidence was found here of the impact of any anthropogenic disturbances on the hydro-chemical status of the study sites, which are directly or indirectly associated with area forms of nature conservation (Poleski National Park and its buffer zone).

To evaluate the habitat quality of the range of values for the physicochemical variables of the groundwater, the results of the analyses should be considered regarding the habitat requirements of the study species. They prefer the same sites on organogenic soils associated with raised or transitional bogs—cold, wet, acidic, oligotrophic, and with moderate light conditions. Due to changes in the hydrological conditions in many places (drainage, irrigation, or regulation), the number of sites of these species as well as the number of individuals at the sites is systematically declining [3,6,18,19,24]. Based on the research, it was established that the values for the physical-chemical variables of the groundwater at the sites of occurrence of the herb species were the same as their habitat preferences. The wide amplitude of values for most of the quality variables of groundwater at the sites of occurrence of the plant species in the peatlands confirmed their broad range of ecological tolerance about the nature of the ecosystem (peatland) to these factors and the regional locations. Differences were found for factors promoting the maintenance of numerous, more abundant populations of a given species at the study sites. In the case of *Comarum palustre*, the values of N-NH_4_ = 0.1–0.21, Ca = 6.8–25.3, Mg = 0.77–2.79, DOC = 25.9–52.9 (mg·dm^−3^), EC = 72.2–107.6 µS· cm^−1^, and pH = 5.3–5.7 were favorable to the occurrence of more abundant populations (Serafin et al. 2022b). For bogbean, higher concentrations of N_tot_ (4.16–27.4 mg·dm^−3^) and P_tot_ (0.93–0.14 mg·dm^−3^) and lower pH (5.23–5.55) and EC (70.4–112 μS·cm^−1^) can be regarded as a set of conditions favoring the normal functioning of individuals of the species, which translates to its abundant occurrence (Serafin et al. 2017). For small cranberry, lower values for these variables, i.e., P_tot_ = 0.17–0.36 mg·dm^−3^, P-PO_4_ = 0.1 mg·dm^−3^, and DOC = 33.81–55.90 mg·dm^−3^, can be considered to favor increased occurrence of the species [18].

Comparative analysis of the sites of occurrence of individual herb species based on the criterion of the community’s abundance of a given species revealed substantial differences despite identical habitat preferences and wide ecological valence. For *Comarum palustre*, the abundant sites were distinguished by lower values for nitrogen fractions, Ca, Mg, pH, EC, and DOC than at the less abundant sites. In the case of *Menyanthes trifoliata*, the abundant sites had higher values for nitrogen fractions, P_tot_, K, and DOC and lower values for Ca, Mg, EC, and pH, while the sites with abundant *Oxycoccus palustris* had higher values for nitrogen fractions, Ca and pH and lower values for P fractions, Na, K, and Mg. Similarly, analysis of the correlations between mean values for physical-chemical variables of the water at the sites, based on the abundance criterion, revealed differences in strong and moderately strong positive and negative correlations depending on the species of herb plant and the type of site. For example, for *Comarum palustre,* strong positive correlations at the more abundant sites were found between N-NH_4_ and P fractions; Mg and K; and EC and Ca, while strong negative correlations were found for pH with N-NH_4_ and with P fractions. For *Menyanthes trifoliata* at such sites, strong positive correlations at the more abundant sites were found for N-NO_2_ with N-NO_3_, SO_2_ with Ca, Na with K, and EC with pH, and strong negative correlations for pH with phosphorus fractions, K and DOC. For *Oxycoccus palustris*, strong positive correlations at abundant sites were recorded for N_tot_ with N-NH_4_, P_tot_, K and DOC; Ca with Mg; and EC with pH, strong negative correlations for P_tot_ and DOC with pH and EC. For sites with low abundance of a given species, there were many strong but only positive correlations between various pairs of variables (the exception was *O. palustris* with a negative correlation for pH with K and N-NH_4_).

Differences between abundant and non-abundant sites of the herb species were also confirmed by the non-parametric Mann–Whitney U test. For the significance level of *p* < 0.05, significant differences in values were obtained for various variables with respect to various species, e.g., N-NH_4_, DOC, Ca, and EC for *C. palustre*, N_tot_, Ca, pH, and EC for *M. trifoliata*, and phosphorus fractions for *O. palustris*. The status of the habitat diversity of the sites is also confirmed by the PCA. Despite the similarities in various key physicochemical variables pairing similar sites rs, e.g., B–K, D–M, and, to a lesser extent, K–BZ, these pairs of sites did not have similar abundance of the species studied. As the study sites are located in an area with potentially limited human impact, these differences confirm the effect of internal metabolism on the hydro-chemical status of peatland ecosystems (according to [6,7,43,45,60]). In the peatlands of Eastern Poland, despite similar habitat requirements, this status had different effects on the characteristics of the occurrence of the herb species. They showed a wide spectrum of ecological tolerance for many of the physical-chemical variables of the groundwater. Their occurrence is, therefore, not the only indicator of habitat changes at the study sites and is most likely determined by other habitat features not analyzed in this study, e.g., the metabolic activity of soil microbes or the species composition of the phytocoenoses [6,7]. Therefore, the occurrence of the species may only slightly reflect habitat changes pertaining to the analyzed variables of the peatbog water.

The above analyses, therefore, confirmed the research hypothesis that in natural conditions groundwater quality determines the occurrence of herb species with the same habitat preferences in varied ways.

Another question is the practical use of data pertaining to the hydro-chemical characteristics favoring the occurrence of the herb species whose acquisition from nature entails difficulties associated with the terrain (wetlands), various forms of nature conservation (legal limitations), or progressive contraction of suitable habitats (human impact).

Organic cultivation preserving the natural physicochemical variables of peatland groundwater for a given species provides the opportunity to obtain high-quality herbal material with a natural composition of biologically active substances [3,18,19]. The present study on the optimal habitat quality factors for the occurrence of selected herb species is an element of pilot research on the concept of a medicinal drug with a natural composition of biologically active ingredients in accordance with WHO recommendations [66].

The natural hydro-chemical habitat characteristics in organic cultivation will make it possible to preserve the natural composition of bioactive substances in the herbal material. The most important stage of the experiment is, thus, hydro-chemical optimization of the substrate in accordance with the analyzed characteristics of natural habitats. This is preceded by laboratory analyses of the medicinal potential of herbal materials, which are currently underway.

## 5. Conclusions

The analysis of the obtained research results allowed us to answer the research questions. It was found that the quality variables of shallow groundwater in the peatlands of the Łęczyńsko-Włodawskie Lake District (selected representative sites) do not indicate increasing human impact or deregulation of hydrological conditions. The natural hydro-chemical characteristics of the peatbogs show fluctuations associated with the dynamics of the internal metabolism of the ecosystem. The analyzed physicochemical factors of the shallow groundwater are within the ranges of habitat preferences for the analyzed herb species. The herb species have wide ecological tolerance for the analyzed physicochemical variables of peatland water.

It was also established that identical habitat preferences do not reflect identical values for physicochemical variables of water essential for building populations of the species and that the occurrence of the plant species is determined by the hydro-chemical characteristics of the habitat, but the characteristics of their occurrence only indicate its hydro-chemical characteristics to a small degree.

## Figures and Tables

**Figure 1 ijerph-20-02788-f001:**
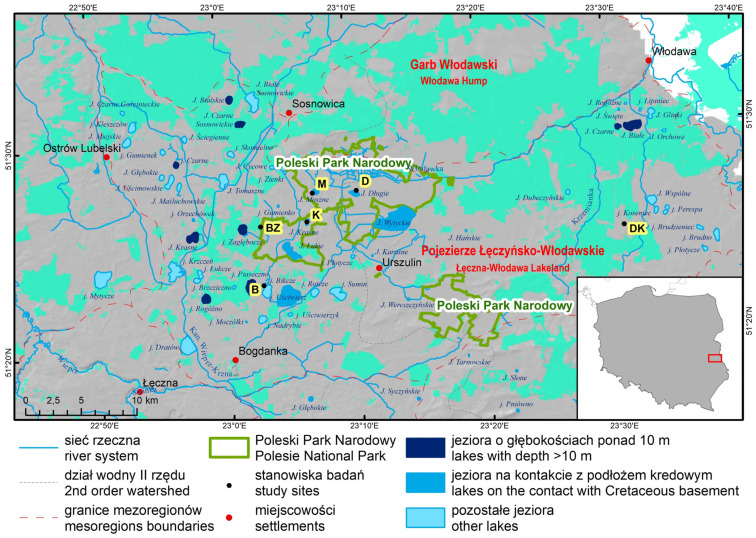
Locations of the study sites in the Łęczyńsko-Włodawskie Lake District; B (Bikcze), D (Długie), K (Karaśne), M (Moszne), BZ (Blizionki), DK (Dekowina) against the background of the relief (gray shade) and the range of forests (green shade).

**Figure 2 ijerph-20-02788-f002:**
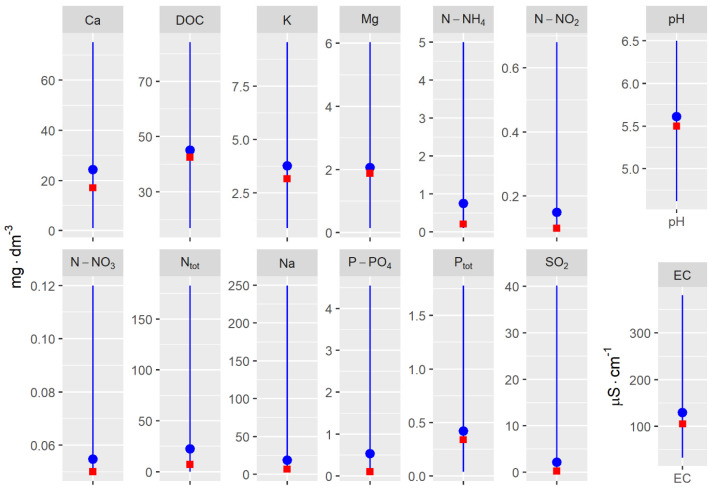
Distribution of values of physicochemical factors of piezometric groundwater at the study sites in 2011–2013. The blue line, blue dot, and red square represent the range of values, mean value, and median, respectively.

**Figure 3 ijerph-20-02788-f003:**
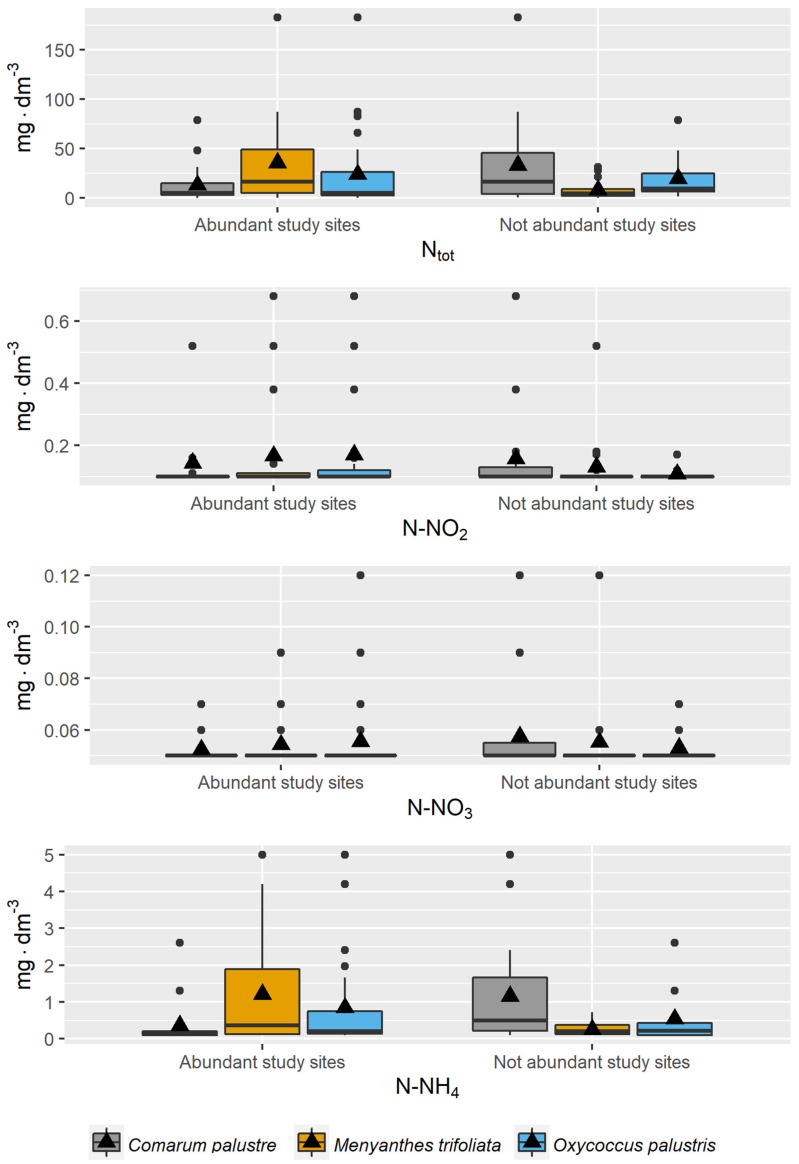
Distribution of values of nitrogen fractions in piezometric groundwater at the study sites for the study species in 2011–2013 (criterion of abundance in habitats). The horizontal line across the central region of the box represents the median. The mean value of the data is marked by a filled triangle. The whiskers are drawn to the most extreme observations. Any observation not included between the whiskers is plotted as an outlier with a filled dot.

**Figure 4 ijerph-20-02788-f004:**
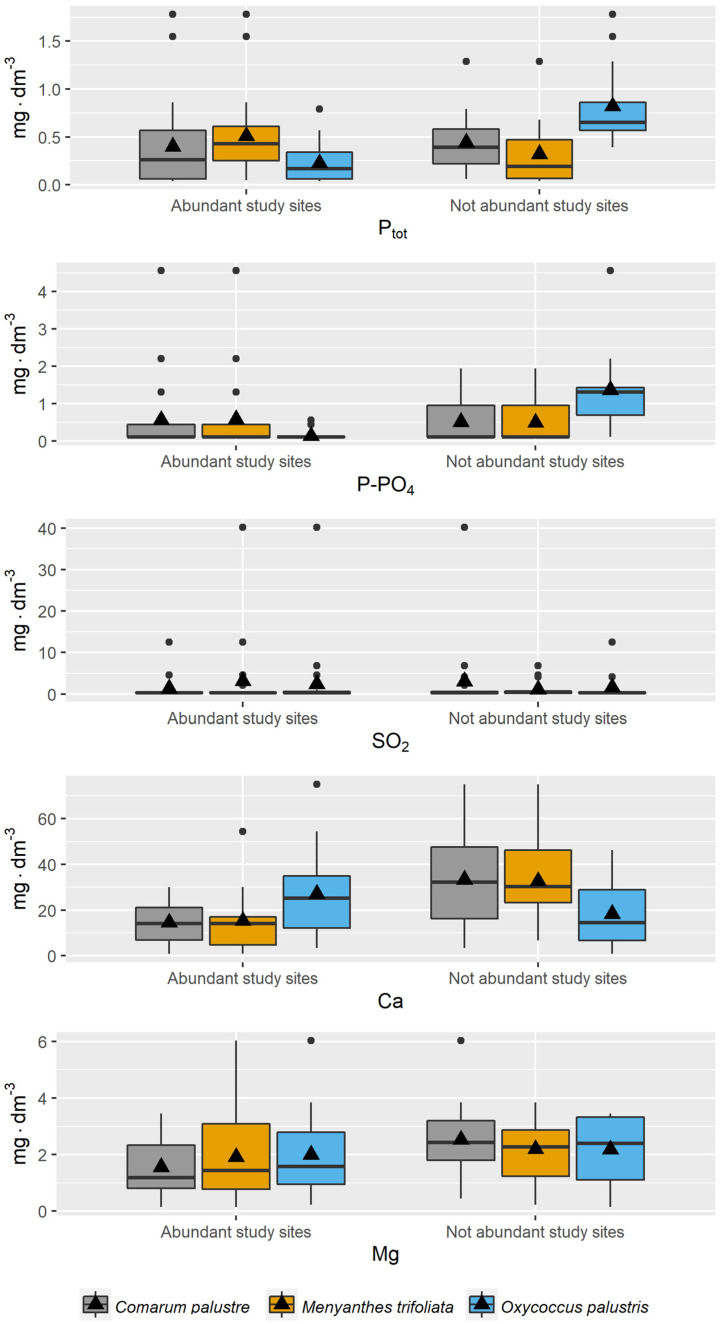
Distribution of values of phosphorus fractions, SO_2_, Ca, and Mg in piezometric groundwater at the study sites for the study species in 2011–2013 (criterion of abundance in habitats).

**Figure 5 ijerph-20-02788-f005:**
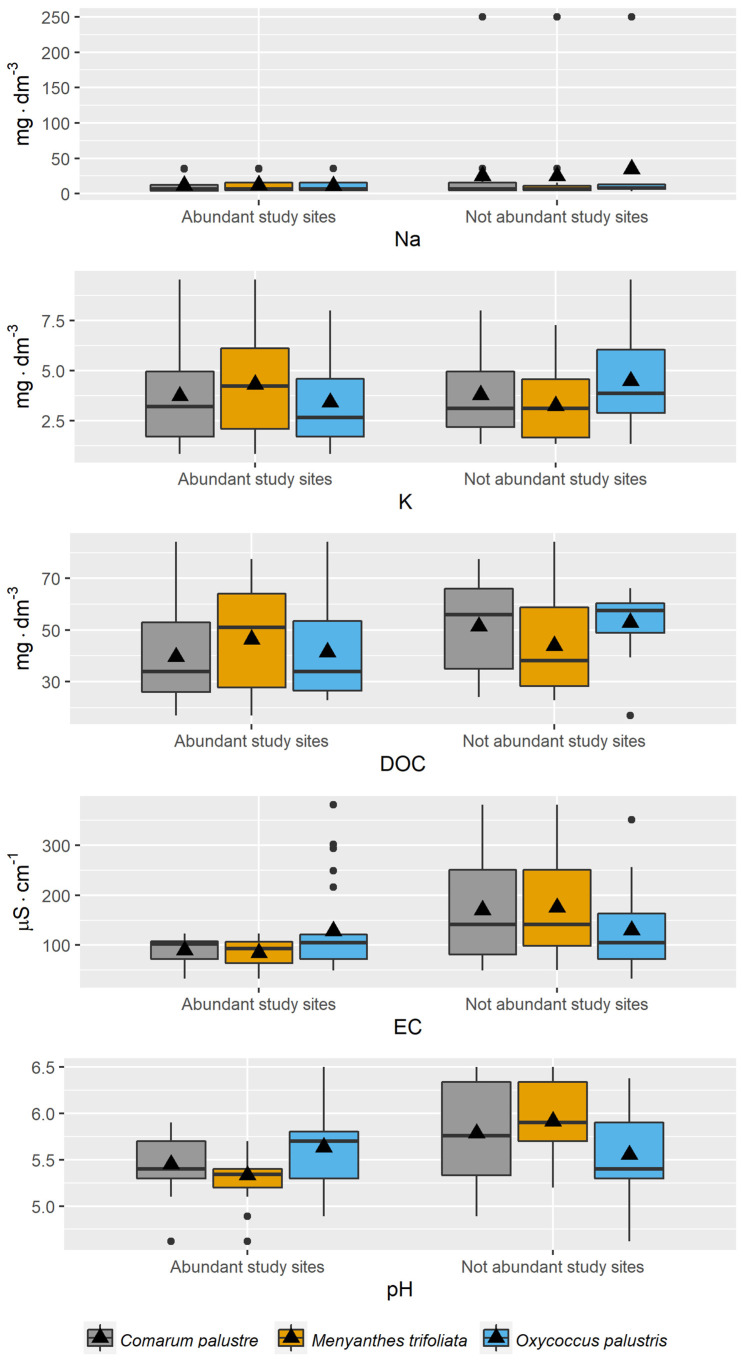
Distribution of Na, K, DOC, EC, and pH values in piezometric groundwater at the study sites for the study species in 2011–2013 (criterion of abundance in habitats).

**Figure 6 ijerph-20-02788-f006:**
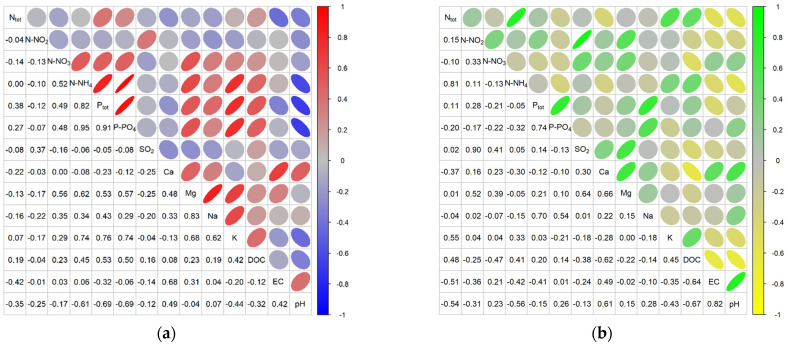
Correlations between physicochemical factors of peatland water as pooled data for *Comarum palustre* in 2011–2013: (**a**) abundant sites, (**b**) non-abundant sites. Ellipse angulation and color intensity are proportional to the Pearson correlation coefficient: positive correlations are shown in red/green and negative correlations in blue/yellow.

**Figure 7 ijerph-20-02788-f007:**
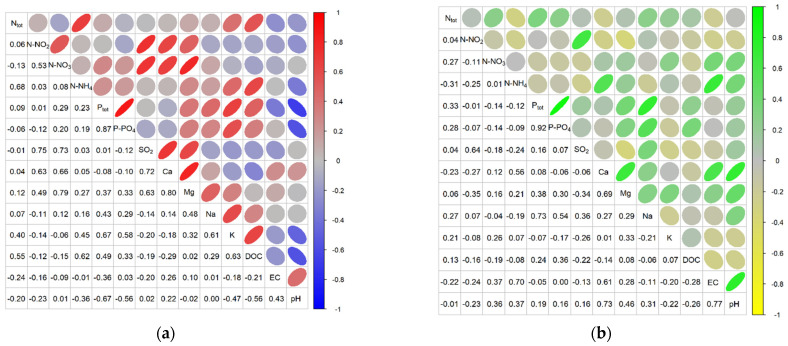
Correlations between physicochemical factors of peatland water as pooled data for *Menyanthes trifoliata* in 2011–2013 (**a**) abundant sites, (**b**) non-abundant sites.

**Figure 8 ijerph-20-02788-f008:**
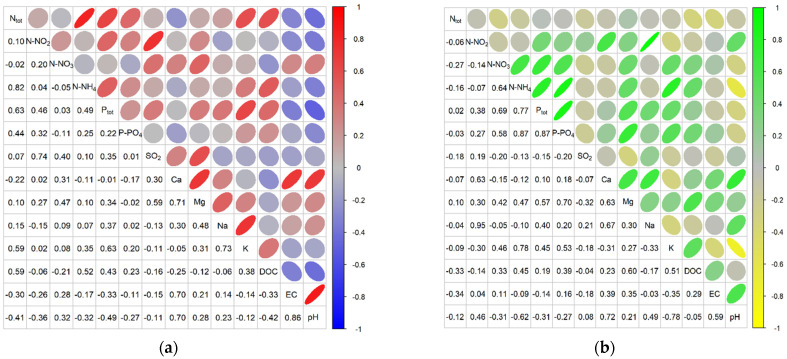
Correlations between physicochemical factors of peatland water as pooled data for *Oxycoccus palustris* in 2011–2013: (**a**) abundant sites, (**b**) non-abundant sites.

**Figure 9 ijerph-20-02788-f009:**
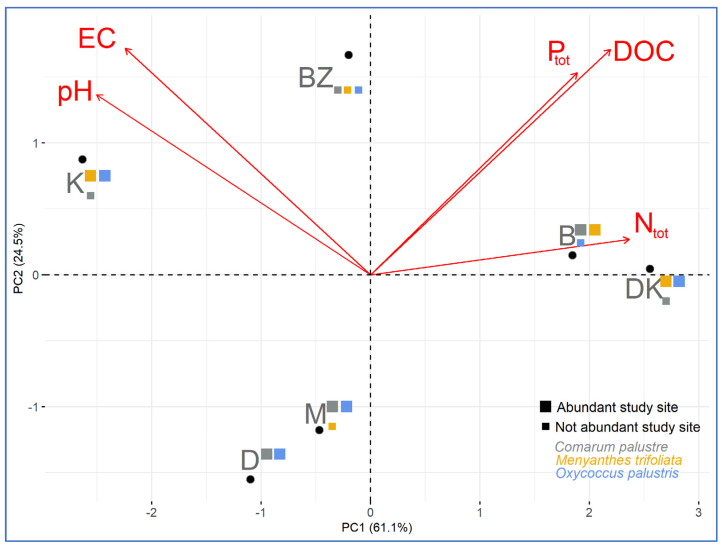
Principal component analysis (PCA) biplot for the physical-chemical variables at the study sites, taking into account the abundance of individuals of the studied species.

**Table 1 ijerph-20-02788-t001:** Mann–Whitney U test results for comparison of physical-chemical variables at study sites with high and low abundance of the study species.

Species	Variable	Abundance of Study Sites	N	Sum of Ranks	Mann-Whitney U Test Results	*p*-Value
*Comarum palustre*	N_tot_	Abundant	21	371		
		Not abundant	19	449	140	0.110
	N-NO_2_	Abundant	21	406		
		Not abundant	19	415	174.5	0.507
	N-NO_3_	Abundant	21	407		
		Not abundant	19	414	175.5	0.524
	N-NH_4_	Abundant	18	231		
		Not abundant	17	399	60	0.002
	P_tot_	Abundant	21	376		
		Not abundant	19	444	145	0.144
	P-PO_4_	Abundant	21	420		
		Not abundant	19	400	189	0.787
	SO_2_	Abundant	20	381		
		Not abundant	19	399	171	0.603
	Ca	Abundant	15	170		
		Not abundant	16	326	50	0.006
	Mg	Abundant	15	191		
		Not abundant	16	305	71	0.055
	Na	Abundant	15	222		
		Not abundant	16	275	101.5	0.477
	K	Abundant	15	238		
		Not abundant	16	258	118	0.953
	DOC	Abundant	21	349		
		Not abundant	19	471	118	0.028
	EC	Abundant	21	359		
		Not abundant	20	502	128	0.033
	pH	Abundant	21	360		
		Not abundant	19	460	129	0.058
*Menyanthes trifoliata*	N_tot_	Abundant	21	517		
		Not abundant	19	303	113	0.019
	N-NO_2_	Abundant	21	446		
		Not abundant	19	374	184	0.685
	N-NO_3_	Abundant	21	429		
		Not abundant	19	391	198	0.978
	N-NH_4_	Abundant	18	368		
		Not abundant	17	262	109	0.151
	P_tot_	Abundant	21	488		
		Not abundant	19	332	142	0.123
	P-PO_4_	Abundant	21	448		
		Not abundant	19	372	182	0.645
	SO_2_	Abundant	20	367		
		Not abundant	19	413	157	0.361
	Ca	Abundant	15	172		
		Not abundant	16	324	52	0.007
	Mg	Abundant	15	214		
		Not abundant	16	282	94	0.313
	Na	Abundant	15	248		
		Not abundant	16	249	112.5	0.782
	K	Abundant	15	263		
		Not abundant	16	233	97	0.374
	DOC	Abundant	21	443		
		Not abundant	19	377	187	0.745
	EC	Abundant	21	316		
		Not abundant	20	545	85	0.001
	pH	Abundant	21	283		
		Not abundant	19	537	52	0.000
*Oxycoccus palustris*	N_tot_	Abundant	27	514		
		Not abundant	13	306	136	0.260
	N-NO_2_	Abundant	27	567		
		Not abundant	13	253	162	0.707
	N-NO_3_	Abundant	27	549		
		Not abundant	13	272	170.5	0.897
	N-NH_4_	Abundant	24	445		
		Not abundant	11	185	119	0.657
	P_tot_	Abundant	27	394		
		Not abundant	13	427	15.5	0.000
	P-PO_4_	Abundant	27	397		
		Not abundant	13	424	18.5	0.000
	SO_2_	Abundant	27	549		
		Not abundant	12	231	153	0.796
	Ca	Abundant	21	366		
		Not abundant	10	130	75	0.213
	Mg	Abundant	21	326		
		Not abundant	10	170	95	0.688
	Na	Abundant	21	323		
		Not abundant	10	173	92	0.597
	K	Abundant	21	311		
		Not abundant	10	185	80	0.301
	DOC	Abundant	27	488		
		Not abundant	13	332	110	0.061
	EC	Abundant	27	570		
		Not abundant	14	291	186	0.945
	pH	Abundant	27	562		
		Not abundant	13	259	167.5	0.829

## Data Availability

Not applicable.

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
