# Peer review of "The Influence of Shallow Peatland Water Quality on Characteristics of the Occurrence of Selected Herb Species in the Peatlands of Eastern Poland"

_ijerph, 2023, doi:10.3390/ijerph20042788_

Round 1

Reviewer 1 Report

The physicochemical parameters of the quality of shallow groundwater in the peatlands of Eastern Poland has important research significance. The authors has done a good research on the chemical properties of plants and water in the study area, and has obtained many beneficial research results. However, there are still some problems in the writing of the paper, and the authors are suggested to improve the current research as much as possible. In particular, how the physical and chemical properties of water in a specific region affect specific plants, and what role these impacts will play in the future environmental governance of the region, etc.

The following questions are put forward for the authors' reference.

1- In the abstract part, it is necessary to first put forward what is the general background of this study.

2- At the end of abstract, the significance and potential value of the study should be emphasized.

3- The introduction should not only focus on a few regions such as Poland, and other regions with similar environmental problems can be appropriately commented.

4- Line68 The research objective of this paper is a little too scattered, and the authors are suggested to make a certain synthesis and induction.

5- Figure 1 is very unclear, and it is difficult to see the surrounding environmental background characteristics. And as a paper on environmental issues, it should first give the local map of the region as the big background. At least, it should reflect the location characteristics of the study area in Poland, and it is not only a linear map of Poland, but also vegetation and terrain information..

6- Line 148 should be 100 square meter.

7- Line 266-Line278 , There seems to be something missing here.

8- The discussion part needs to study the significance of this study for plant ecological environment.

9- The format of the conclusion should be consistent with the full text.

Author Response

We would like to thank the Reviewer for a thorough review supported by an insightful analysis of the content and form of this paper. It will certainly contribute to the improvement of our scientific and editorial workshop. A great majority of the suggested corrections was introduced (please see new pdf version), and additional doubts we will try to explain below in accordance with the order of suggestions in the review of our paper:

  1. As part of the abstract, we are formally obliged not to exceed a specific number of characters (200). Outlining the background is at least 2-3 sentences, and we have already fully filled the limit. The reviewer did not indicate unnecessary content in the abstract, therefore supplementing it with more complete information is significantly difficult for formal reasons. We know from experience that the essence of an article is the quintessence of information, because the vast majority of recipients read only the abstract. Developing preliminary, supplementary and summary information at the expense of essential information (aim, method, result, conclusion), which is related to the character limit, is in our opinion problematic in terms of substance, although if there were formal possibilities - valuable for the article.
  2. For the same reason, we must treat additional information about the research potential of the results as default for interested readers in the context of the analysis of the conclusions presented in the summary or as requiring reading the entire article.
  3. Habitat features of most of the regions analyzed by various scientists outweigh the overwhelming influence of climate, therefore comparing different research locations in different physiographic units is quite unjustified. The same species in peat bogs of Northern Poland grow in slightly different physical and chemical conditions than in Eastern Poland, despite the same ecological requirements. They often create local varieties - ecotypes. Of course, there are even greater differences between different countries in Europe and the world. The thematic approach presented in our article is also related to the perspective of organic cultivation of these species in the study area, based on the material of individuals obtained from the research sites (in vitro breeding). However, the inclusion of information about the same species found in other regions in the context of shallow groundwater quality is our next research goal. For the above reasons, please release us from the need to supplement the information with the content proposed by the Reviewer.
  4. We do not really understand the Reviewer's suggestion on this point. The aim of the study was synthetically described (Line 64-67 of the original version of the manuscript). Problems understanding this information may be due to editorial errors (misplaced bullets), which we have already corrected. Additional information is: research hypothesis and research questions that allow you to finally formulate conclusions, so they are not the goal of research in itself. The practical goal is only a long-term consequence of research and conclusions, it should not be combined with the main goal, therefore it is presented in accordance with the logical composition at the end of this paragraph.
  5. As suggested by the Reviewer, we have improved Figure 1 by enriching it with the proposed content.
  6. We have corrected this bug as suggested. This also results from the editorial transformation of our article into a journal format.
  7. Fortunately, this is just an editorial error (see point 6) and not missing text, which continues on line 279. We corrected this mistake.
  8. At this point, we also have some doubts as to the intentions of the Reviewer. In our opinion, 90% of discussions refer to the issues indicated by the Reviewer. They concern the analysis of the obtained values of physical and chemical factors in terms of habitat characteristics of the environment of the occurrence of the studied plant species. We described the role of selected physico-chemical factors in the environment and their informative nature regarding the impact of anthropopressure, and therefore valuable in connection with the analysis of environmental impacts on the habitat. We did not record significant anthropogenic impacts during the research period, so we treat them as natural. Their values may fluctuate due to the metabolism of the peat bog, which we have also indicated. The issues of botanical analyzes are the subject of our subsequent works. We do not know what ecological aspects we should therefore additionally refer to. We hope these comments will address the suggestion in the review.
  9. We adapted the applications to the descriptive format as suggested by the Reviewer.

We hope our supplementations and adjusting the form of publication to required standards satisfies the Reviewer and the Editorial Collegium of the journal, and allows for publication of the paper in International Journal of Environmental Research and Public Health.

With best regards,

The author’s collegium

Reviewer 2 Report

This manuscript has the potential to be published in IJERPH, but it needs substantial improvements in the methodology and presentation of results. Suggestions for improvement are presented in the attached file.

Author Response

We would like to thank the Reviewer for a thorough review supported by an insightful analysis of the content and form of this paper. It will certainly contribute to the improvement of our scientific and editorial workshop.

Thank you for your remarks and comments in the pdf version of the article. We referred to all of the Reviewer's indications and, in the vast majority, made additions in accordance with the suggestions. We tried to explain the omitted corrections in the comments (in new .doc version).

We hope our supplementations and adjusting the form of publication to required standards satisfies the Reviewer and the Editorial Collegium of the journal, and allows for publication of the paper in International Journal of Environmental Research and Public Health.

With best regards,

The author’s collegium 

Reviewer 3 Report

The reviewer supposes that among the parameters determining the development of herbs under study must be some parameters which were not determined by the aythors. They have included into their list just standard set of parameters and observed that they are not of importance for these herbs in this given location. The reviewer supposes the manuscript reflects the study which is not completed.

Author Response

We would like to thank the Reviewer for a thorough review supported by an insightful analysis of the content and form of this paper. It will certainly contribute to the improvement of our scientific and editorial workshop.

However, as a group of scientists studying environmental impacts on the occurrence status of many peatland plant species for almost 30 years, we are somewhat surprised by this interpretation of our research. We are a bit disappointed that the Reviewer did not indicate what physical and chemical factors of peat bog waters should be additionally taken into account, but only made assumptions that there may be others - important for plant development (although we did not study the development of plants, only their occurrence), which we did not take into account . Our long-term observations show that in areas associated with spatial forms of protection, we can analyze the possible, minor impact of anthropogenic pressure on the habitat status of many plant species by examining the water quality parameters just proposed. Standard? Yes, because in the examined location there are no specific anthropogenic influences: industrial plants, railway transport lines, etc., which could force research in a different direction, e.g. concentration of heavy metals or radionuclides. We took into account the basic parameters of anthropopressure related to the study area, which is reflected in the Discussion chapter. We treat other parameters as local, natural and non-limiting for the occurrence of the studied plants, in accordance with the adopted published methodology of ecological research in peat bog areas. Similarly, the selected parameters are closely related to the habitat analysis in relation to the preferences of selected herb species. How could we analyze these issues by examining other factors?

Similarly, the statement that our work shows that the tested parameters do not affect the occurrence of individual species of herbs raises similar doubts. The essence of our work is the completely opposite statement. The studied parameters may favor the occurrence of the studied species in different ways despite their identical habitat preferences (Chapter Discussion, Line 429-446, in primary version; Conclusions)

The reviewer's general statement is, in our opinion, rather inadequate to the content of the article, therefore it does not give us any room for further discussion, as there are no more precise comments and more specific accusations to be accepted, or polemics. Of course, one should agree with the statement that no scientific work is finally finished. But at some point we have to make a point. In our opinion, the presented content corresponds to the assumed goals and allows for the formulation of valuable long-term conclusions. The more so that it is another work in the series successfully published in various MDPI journals based on the same methodology. Therefore, it is difficult for us to agree with such a problematic opinion of the Reviewer.

We hope our supplementations and adjusting the form of publication to required standards, in spite of all, satisfies the Reviewer (please see new pdf version) and the Editorial Collegium of the journal, and allows for publication of the paper in International Journal of Environmental Research and Public Health.

With best regards,

The author’s collegium

Round 2

Reviewer 1 Report

The authors have made good revisions and the manuscript has been improved. I think the manuscript has reached the status of publication.

Author Response

We would like to thank the Reviewer for the second round of review and thank you very much for approving our corrections and recommending the work for publication in International Journal of Environmental Research and Public Health.

With best regards,

The author’s collegium

Reviewer 2 Report

I consider that this new version has improved substantially, in comparison with the previous version. This is thanks to the fact that the recommendations of the reviewers were considered.

I strongly recommend making three changes that I requested in the first review:

1. Replace the use of the word “parameter” by that of “variable”. This is because the authors used non-parametric statistical techniques. The use of “parameter” can cause confusion for readers, since in statistics, it is used when the variables meet the assumptions of normality and equality of variances and, therefore, are analyzed with parametric techniques.

2. All information regarding the medicinal properties of the plant species should be removed from the study. This is requested because the work is ecological, not ethnobotanical or ethnoecological.

3. Discuss more fully how the authors verified the human impact on the ecosystem because to achieve this, they needed to compare protected areas versus impacted areas, but the authors only worked in protected areas.

Author Response

We would like to thank the Reviewer for a thorough review supported by an insightful analysis of the content and form of this paper in the second round of review.

Additional doubts we will try to explain below in accordance with the order of suggestions in the review of our paper:

  1. Respecting the Reviewer's view, we have introduced the suggested change, although we have encountered such a suggestion in our scientific works for the first time.
  2. Similarly, in the case of the above issues, we have removed information regarding the medicinal properties of herbs. In our opinion, placing such information would emphasize the purposefulness of the selection of research objects. This is of particular importance in the era of the ecologisation life and competently referred to the character of the magazine (public health). However, we respect the Reviewer's view, which is why we have followed this suggestion.
  3. Based on the explanation below, we've added the appropriate information in the Discussion section.

However, the conclusion of the Reviewer is not obvious. The impact of anthropopressure does not have to be determined only as a result of the suggested comparison.

The selection of research plots as representative was specified in 2010, having previously studied these areas since 1995. In numerous studies, we analyzed habitat characteristics, including water hydrochemistry, soil microbiology and biocenosis of phytocenoses for various groups of plants (boreal relics, herbs). These analyzes made it possible to identify research locations representative for Western Polesie for future field research (raised and transitional bogs). These were semi-natural places with a potentially limited impact of anthropopressure due to the protected areas and their buffer zone, and with regulated hydrological conditions (verification based on scientific information from the Polesie National Park).

The first stage of the work related to this article was therefore to confirm whether there is in fact no impact of anthropopressure in the studied locations, or whether it is insignificant. The key to analyzing this element is the so-called back-effect analysis, often used in ecology. For example, if the pH value of water in a transitional mire is within the typological ranges corresponding to this type of habitat, there is no evidence of a significant impact of anthropogenic pressure. If the species composition and plant communities are typical of a transitional mire, there is no evidence of anthropopressure (indices of flora synanthropization and index numbers of higher plants are used here). If the value of other physico-chemical factors of waters is typical for a given habitat, then there is also no evidence of increased anthropopressure. The greater part of the Discussion chapter is devoted to such a retrospective analysis of the effect, which gives us an obvious scientific answer about the lack or negligible impact of anthropopressure on the development of biocoenosis. Therefore, it confirms the naturalness of the locations selected for research, giving rise to further analyzes of the values of factors that are favorable or less favorable to larger populations.

We sincerely hope that these explains, corrections and additions this time will fully satisfy the Reviewer's expectations and will allow the publication of this article in International Journal of Environmental Research and Public Health.

With best regards,

The author’s collegium

Reviewer 3 Report

Dear authors - every scientific investigation, every observation and every measurement can and must be published. The speculation on them can serve as way to wrong or false conclusions, but the facts themselves must be accessible to other researchers. So, you received the facts, you have the right to propose your own explanation for them. Other researchers may have their own vision of the problem - that is their right. But the facts must be published and the community must have an access to them. Thank you for your job.

Author Response

We would like to warmly thank the Reviewer for recognizing our arguments and recommending this article for publication in International Journal of Environmental Research and Public Health.

 We hope that the final version of the article, after the second round of review, will fully meet the Reviewer's expectations.

With best regards,

The author’s collegium